# Continual Learning in Open-vocabulary Classification with Complementary Memory Systems

**Zhen Zhu**                                                          *zhenzhu4@illinois.edu*
*Siebel School of Computing and Data Science*
*University of Illinois Urbana-Champaign*

**Weijie Lyu**[*]                                                     *wlyu3@ucmerced.edu*
*University of California Merced*

**Yao Xiao**                                                          *yaox11@illinois.edu*
*Siebel School of Computing and Data Science*
*University of Illinois Urbana-Champaign*

**Derek Hoiem**                                                       *dhoiem@illinois.edu*
*Siebel School of Computing and Data Science*
*University of Illinois Urbana-Champaign*

**Reviewed on OpenReview:** *https://openreview.net/forum?id=6j5M75iK3a&noteId=0ABvrBz6Rg*

## Abstract

We introduce a method for flexible and efficient continual learning in open-vocabulary image classification, drawing inspiration from the complementary learning systems observed in human cognition. Specifically, we propose to combine predictions from a CLIP zero-shot model and the exemplar-based model, using the zero-shot estimated probability that a sample's class is within the exemplar classes. We also propose a "tree probe" method, an adaption of lazy learning principles, which enables fast learning from new examples with competitive accuracy to batch-trained linear models. We test in data incremental, class incremental, and task incremental settings, as well as ability to perform flexible inference on varying subsets of zero-shot and learned categories. Our proposed method achieves a good balance of learning speed, target task effectiveness, and zero-shot effectiveness. Code is available at https://github.com/jessemelpolio/TreeProbe.

## 1 Introduction

A major machine learning goal is to create flexible learning systems, which we investigate in the context of image classification, with the following goals:

- Flexible inference: classify an image within any label set, testable via zero-shot classification
- Continual improvement: accuracy improves as new data is received for related tasks without detriment to previous tasks
- Efficient incremental learning: can efficiently update the model with new training examples

Flexible inference and continual improvement lead to robust and widely usable systems. Efficient incremental learning reduces cost of training and facilitates responsive and interactive learning approaches. Imagine interacting with a nature classification app: it begins with open-vocabulary classifiers, providing reasonable predictions for plant and animal categories. Whenever an expert user correct mistakes, the app learns from their input to improve on the labeled species without detriment to its existing capabilities, as shown in Fig. 1.

---

[*]Co-first author. Work is done while being a student at UIUC.

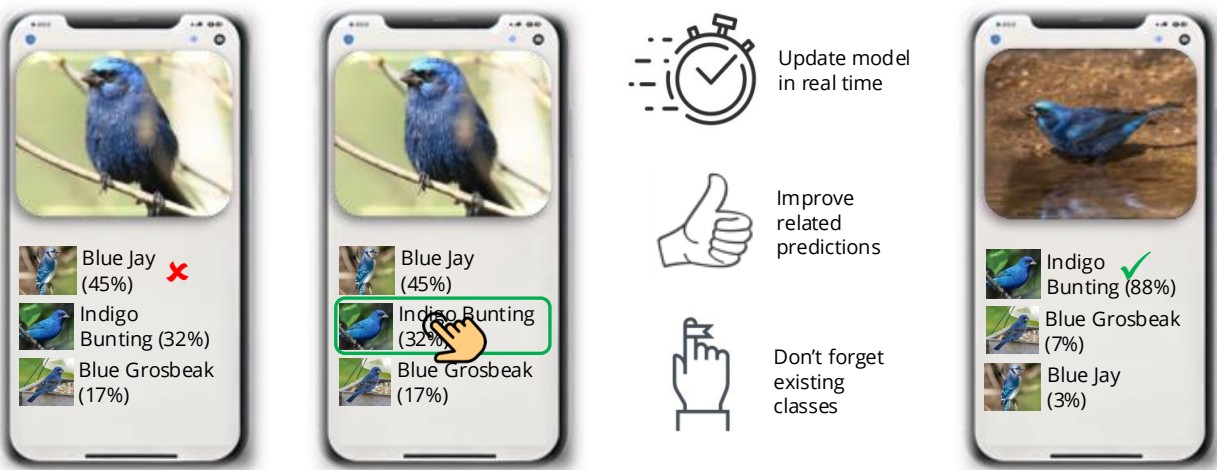

Figure 1: Our work proposes a method to continually expand and improve classification ability, updating the model quickly with each new labeled example. This is especially helpful in long-tailed classification problems, like the depicted nature classification app. Given a user-provided image, the system predicts the likely classes and immediately updates given any corrections. In this way, the system becomes increasingly capable without any costly offline retraining.

However, to our knowledge, no solution exists that meets all these goals. Open vocabulary methods, such as CLIP (Radford et al., 2021), attain flexible inference but cannot easily learn and improve with new examples. Continual learning methods, such as LwF (Li & Hoiem, 2018), iCaRL (Rebuffi et al., 2017), EWC (Kirkpatrick et al., 2016), and GDumb (Prabhu et al., 2020), lack flexible inference and are not easily extended to open vocabulary models. For example, LwF requires maintaining classification heads for previous tasks, and GDumb maintains examples for every possible label; neither strategy directly applies when the label set is unbounded. Recent prompt-based approaches Wang et al. (2022c;b;a); Smith et al. (2023) share the same disadvantages. In fact, one homogeneous system cannot easily accommodate all goals. A deep network can aggregate information over enormous datasets to achieve flexible inference, but updates from small targeted subsets can disrupt other tasks. Instance-based methods can incrementally learn and continually improve, but cannot generalize to novel labels.

To build such a flexible learning system, we are inspired by the flexibility of human learning and inference, enabled by complementary learning systems (CLS) (O'Reilly et al., 2014). The exemplar-based system forms new memories and associations with sparse connections, enabling non-disruptive storage of individual experiences. The consolidated system gradually consolidates information stored in the exemplar-based system to a densely connected network, encoding general knowledge and regularities across experiences. The interplay between these two complementary systems enables the brain to balance the rapid encoding of novel information and gradual generalization of knowledge. How can we make computer learning systems that likewise benefit from both consolidated and exemplar-based memory systems to continually learn with zero-shot inference ability?

We investigate in the context of open-vocabulary image classification, using frozen CLIP as the consolidated system. Our exemplar-based system stores individual representations of learning exemplars, and the challenge is to learn from them in a way that is performant in both accuracy and learning time (Sec. 3.2). To balance speed and performance, we consider classic learning approaches. One option is linear probe, which can achieve good accuracy but learns from a new example in $\mathcal{O}(n)$ time, where $n$ represents the total number of exemplars. K-nearest neighbor (KNN) can learn in $\mathcal{O}(1)$ time but tends to be less accurate than linear probe. Based on local linear models from the lazy learning literature (Bontempi et al., 1999), we propose a "tree probe" method that hierarchically clusters examples and trains linear models for each cluster. The time to learn from a new example is $\mathcal{O}(\log n)$ (in practice, essentially constant time), and the accuracy is close to linear probe.

A second challenge is to predict using both CLIP and exemplar models (Sec. 3.3) to achieve good performance on new tasks while maintaining zero-shot performance. The exemplar model tends to perform well for test samples with labels that are in the exemplar set ("exemplar covered"), while the CLIP model can potentially predict any label. At test time, we may not know whether the label of a given test image is exemplar-covered. Our idea is to use CLIP to estimate the probability that an image's label is exemplar-covered and then use that probability to weight the predictions of the two models.

Our unified system achieves good accuracy, efficient learning, and flexible inference, which meets all three goals for flexible learning systems. We evaluate our method in the forms of data-incremental, class-incremental, and task-incremental learning, as well as flexible inference with categorization tasks that involve some, all, or none of the exemplar-covered labels. Additionally, we compare on an existing benchmark (Zheng et al., 2023), outperforming all compared approaches with the additional benefit of efficient learning. In summary, the reader may benefit from the following **paper contributions**:

- Tree-probe exemplar model: Our locally linear models using a hierarchical clustering can be considered a member of the long-studied lazy learning approaches (Bontempi et al., 1999), but we are not aware of this specific method being proposed or used. Tree-probe has practically constant training time in number of training samples and achieves better accuracy than KNN approaches. This is attractive for interactive search, active learning, and other applications where annotated examples are received in a trickle and fast learning is required.
- Exemplar and consolidated model combination with embeddings: Our approach, to use the consolidated model to estimate applicability of the exemplar model and to combine model predictions in the label embedding space, enables effective continual open-vocabulary learning and performs significantly better than alternatives we tested.
- Flexible learning/inference: Our proposed new benchmark evaluates both the ability to continually learn from new samples in various learning scenarios and to flexibly apply those learnings to various category sets. This may provide a useful test framework to further improve open-vocabulary continual learning.

## 2 Related works

### 2.1 Open-vocabulary image classification

Open-vocabulary image classification aims to categorize images without constraints of predefined labels. CLIP (Radford et al., 2021) achieves this by learning embedding spaces of images and text, trained via a contrastive objective that corresponding image/text pairs will be more similar than non-corresponding pairs. A new image can be classified into an arbitrary set of labels based on the similarity of the image embedding features to the text embedding features corresponding to each label. An alternative approach is to jointly encode the image and text information and decode into text. Following this approach, vision-language models, such as GPV-1 (Gupta et al., 2022), GPV-2 (Kamath et al., 2022), VL-BERT (Su et al., 2020), VL-T5 (Cho et al., 2021), Unified-IO (Lu et al., 2022), Gato (Reed et al., 2022), and Flamingo (Alayrac et al., 2022), can solve a broad range of tasks that includes open-vocabulary image classification, but typically are larger and more complex than CLIP. We use CLIP as our consolidated model, due to its simplicity and verified effectiveness in a broad range of tasks (Wortsman et al., 2022b; Gu et al., 2021; Ghiasi et al., 2021; Lin et al., 2022). According to the data overlap analysis in (Radford et al., 2021), though trained on Internet-scale data, CLIP under-performs models trained for particular benchmarks. Our method continually and efficiently improves CLIP's capability for new concepts, enabling users to add customized expertise while maintaining general capability.

### 2.2 Continual learning

Wang et al. (2023) provides a comprehensive survey of continual learning techniques, which aim to acquire new knowledge over time while preserving previously learned knowledge (McCloskey & Cohen, 1989). Approaches to continual learning can be broadly categorized into regularization (Li & Hoiem, 2016; Kirkpatrick et al., 2016; Zenke et al., 2017), parameter isolation (Aljundi et al., 2017; Rusu et al., 2016; Serrà et al., 2018;

Mallya & Lazebnik, 2018; Zhang et al., 2020), and rehearsal methods (Rebuffi et al., 2017; Yan et al., 2021; Lopez-Paz & Ranzato, 2017; Shin et al., 2017). Regularization techniques generally impose constraints on the learning process to alleviate forgetting. Parameter isolation methods maintain learning stability by fixing subsets of parameters (Mallya & Lazebnik, 2018; Serrà et al., 2018) or extending model with new parameters (Rusu et al., 2016; Yoon et al., 2017; Zhang et al., 2020). Recently, inspired from the Prompt Tuning technique (Lester et al., 2021), prompt-based approaches (Wang et al., 2022c;b;a; Smith et al., 2023; Samadh et al., 2023) continually learn task-specific prompts to condition frozen ViTs (Dosovitskiy et al., 2021) for image classification. The learned prompts can be considered as expanded parameters of task-specific knowledge but tend to overfit as the training progresses, therefore losing the generalization capability of the frozen model. Khattak et al. (2023) propose to use a self-regulating mechanism to learn both task-specific and task-agnostic prompts but the more effective deep prompting technique requires appending prompts at each transformer layer, impacting the training efficiency. Rehearsal methods involve storing and replaying past data samples during training (Shin et al., 2017; Bang et al., 2021; Zhang et al., 2022). Some approaches (Tiwari et al., 2022; Yoon et al., 2022; Hao et al., 2023) consider to maintain a coreset when concerning the memory capacity, but simple sampling methods work better if memory capacity is not a concern (Prabhu et al., 2023).

Many approaches highlight the effectiveness of incorporating neuro-inspired adaptability to balance memory stability and learning plasticity in artificial intelligence systems (Parisi et al., 2018; Kemker & Kanan, 2018; Ayub & Wagner, 2023; Jung et al., 2023; Liang et al., 2023; Madaan et al., 2023) For instance, the CLS theory is exemplified in DualNet (Pham et al., 2021), which integrates supervised and self-supervised learning to facilitate robust continual learning by emulating the hippocampus and neocortex functions. Similarly, Sarfraz et al. (2023) uses sparse coding techniques to mimic the brain's memory retention capabilities, and Parisi et al. (2018) leverages dual-memory recurrent self-organization for spatiotemporal representation. CBCL-PR (Ayub & Wagner, 2023) applies cognitively inspired models to improve the class-incremental learning in robotics. A recent advancement (Khan et al., 2024) leverages a mixture-of-adapters architecture with generative routing to mitigate forgetting while maintaining parameter efficiency. FearNet (Kemker & Kanan, 2018) creates a brain-inspired dual-memory system containing a hippocampal network for recent memories and a prefrontal cortex network for long-term storage, and a basolateral amygdala network for model selection. We share the inspiration from CLS theory, but our work is distinguished in its focus on efficient incorporation of new examples, ability for open vocabulary prediction, and demonstration on larger scale datasets.

To generalize to unseen categories, a line of methods (Skorokhodov & Elhoseiny, 2021; Gautam et al., 2021) continually learn attributes (Farhadi et al., 2009; Lampert et al., 2014) and use attribute-class relations as a way to generalize from seen to unseen categories. Despite using extra attribute annotations, these methods usually under-perform on unseen categories as compared to CLIP (Radford et al., 2021). We are not aware of many works that address continual learning in the context of open-vocabulary image classification. WiSE-FT (Wortsman et al., 2022a) finetunes CLIP encoders on target tasks and averages finetuned weights with original weights for robustness to distribution shifts. Likewise, CLS-ER (Arani et al., 2022) exponentially averages model weights in different paces for its plastic and stable model to balance learning and forgetting, drawing inspirations from CLS. Though not specifically designed for open-vocabulary continual learning, WiSE-FT and CLS-ER can be modified to counter forgetting. Recently, ZSCL (Zheng et al., 2023) explores this setting but mainly focuses on finetuning consolidated CLIP encoders. It combines LwF (Li & Hoiem, 2016) and a weight ensemble idea similar to WiSE-FT as the remedy to reduce forgetting during finetuning. Rather than refining or expanding a single model, our approach is to create a separate exemplar-based model that complements a pre-trained and non-updated more general consolidated model. In our tree-probe method, the exemplar-based model continually learns using a local rehearsal, such that incorporating a new example requires only storing the new image and text embeddings and tuning a linear model using a subset of visually similar examples. This requires little storage or computation and preserves the generality of the consolidated model.

## 2.3 Instance-based learning

Instance-based learning (IBL) (Aggarwal, 2014) is a family of learning algorithms that construct a decision boundary using a memory of training instances, allowing for efficient and flexible adaptation to new data points. This learning paradigm relies on the principle of local approximation, where predictions are made

based on the stored instances that are most similar to the query. One of the most well-known IBL methods is the *k*-Nearest Neighbors (KNN) algorithm, which has been extensively studied for its simplicity and effectiveness in various domains, including classification and regression tasks (Guo et al., 2003; Zhang et al., 2006; 2017; Song et al., 2017).

Closely related to IBL is *lazy learning*, in which training data is organized for prediction at inference time instead of training time. This approach can be more practical than batch or "eager" learning in applications like online recommendation systems, when the training data is constantly evolving. Lazy learning approaches can include KNN or locally RBF network (Bottou & Vapnik, 1992), linear regression (Atkeson et al., 1997) or classification models (Aggarwal, 2014), where models are trained based on neighbors to the query.

We investigate KNN, a form of locally linear classifiers, and globally linear classifiers, exploring their trade-offs for training and inference time and accuracy on a growing set of exemplars, as well as how to combine their predictions with a static image-language foundation model.

## 3  Method

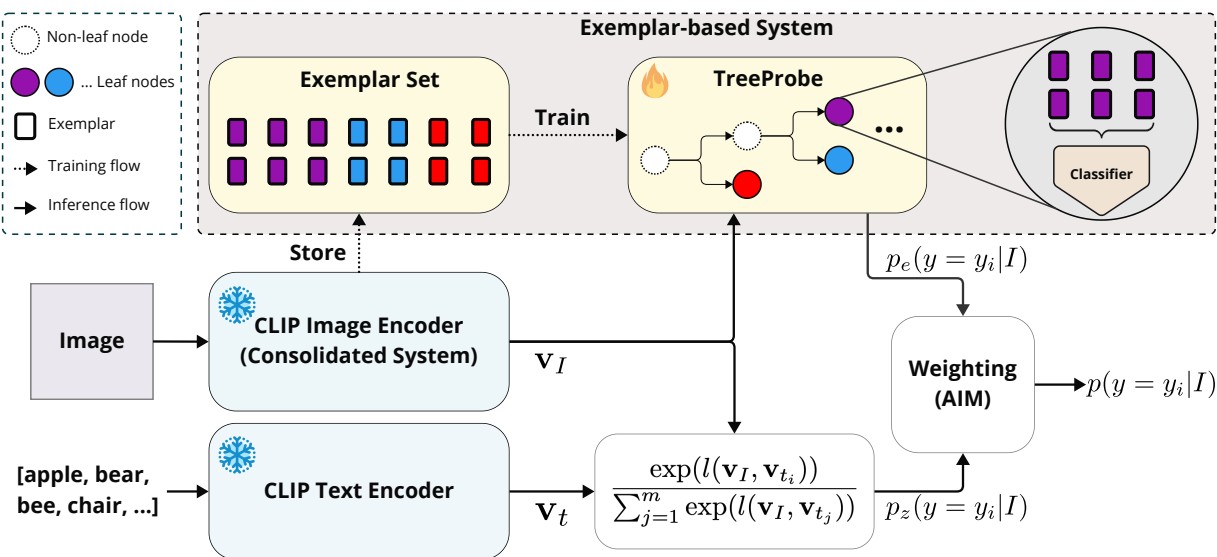

Figure 2: Method overview. (a) Our model integrates exemplar-based and consolidated systems. Final results are made by fusing the predictions from both systems using a weighting method such as AIM. (b) Illustration of TreeProbe, which incrementally adds and hierarchically clusters examples. TreeProbe trains logistic regression classifiers using examples in updated leaf nodes. Colors of exemplars indicate different categories.

Given an image $I$ and label set $Y$, our system's inference task is to assign the correct label $y \in Y$. Each training example (or "exemplar") consists of an image, and a ground-truth label. Our problem setting is called "open-vocabulary" or "zero-shot", as the labels can be represented and related through text, and the task at inference time may contain labels or candidate label sets not observed during training. The training goal is to efficiently update the model with each new example, such that accuracy improves for related labels while maintaining zero-shot performance.

Analogous to complimentary learning systems (CLS) theory (O'Reilly et al., 2014), our approach (Fig. 2) uses two models: a CLIP image/text encoder as the consolidated system (Sec. 3.1), and an exemplar-based system that stores and encodes image and label embeddings (Sec. 3.2). The CLIP model can assign confidence to any label from an arbitrary set. The exemplar-based model can acquire expertise that complements the CLIP consolidated model, but its scope is limited to exemplar labels. In Sec. 3.3, we propose simple mechanisms to combine the predictions of each model to retain zero-shot capability while improving in tasks related to received examples.

### 3.1 Consolidated system

Deep networks, such as CLIP (Radford et al., 2021), relate to consolidated systems in humans in that they repeatedly process examples to iteratively consolidate experience and improve predictive ability in dense weight connections. We use CLIP as the consolidated system, taking advantage of its representations and open-vocabulary predictive ability learned from batch training on massive datasets. We wish to enable fast learning and retain zero-shot ability; therefore, we do not retrain or fine-tune the CLIP encoders. Adapters and prompt-based methods cannot meet the goal of efficient training because they require relatively large computation to run through a part or the whole CLIP encoders. Instead, we train complementary exemplar-based models to be used in combination with CLIP.

CLIP encodes an input image $I$ using an image encoder $f_{\text{img}}$ to an image embedding $\mathbf{v}_I = f_{\text{img}}(I)$, and encodes an input collection of text labels $T = t_1, t_2, \ldots, t_m$ using a text encoder $f_{\text{txt}}$ to text embeddings $\mathbf{v}_{t_i} = f_{\text{txt}}(t_i)$. Following (Radford et al., 2021), a text label $t_i$ is framed as a sentence or caption contextualizing the image label $y_i$, such as "a photo of a F-16A/B, a type of aircraft" to represent the image label "F-16A/B" in an airplane classification task. The model computes logits for each label as cosine similarity between the image and text label, weighted by temperature $\tau$ (=100 in CLIP):

$$l(\mathbf{v}_I, \mathbf{v}_{t_i}) = \tau \cdot \frac{\mathbf{v}_I \cdot \mathbf{v}_{t_i}}{|v_I||v_{t_i}|}. \tag{1}$$

Logits can be converted to probabilities using a softmax function:

$$p_z(y = y_i | I) = \frac{\exp(l(\mathbf{v}_I, \mathbf{v}_{t_i}))}{\sum_{j=1}^{m} \exp(l(\mathbf{v}_I, \mathbf{v}_{t_j}))}. \tag{2}$$

The label with maximum probability is predicted.

### 3.2 Exemplar-based system

For the exemplar-based system, given one or more exemplars, our goal is to maximize classification performance with minimal training time and acceptable inference time. We consider two approaches: instance-based and model-based prediction. Instance-based prediction leverages the exemplar set directly by retrieving and comparing samples. Model-based prediction seeks to capture the underlying structure of the data through a parameterized model. In both cases, our approach leverages the embeddings from CLIP encoders to reduce storage and improve sample efficiency of learning.

Our exemplar memory module $M$ stores encoded image embeddings and their labels. Each entry in the memory can be denoted by $M_j = \{\mathbf{v}_{I_j}, y_j\}$ where $j$ represents the entry index, $\mathbf{v}_{I_j}$ represents the image embedding of $I_j$, and $y_j$ is the corresponding label.

Given a set of target labels, the exemplar-based model can produce a probability for each label $p_e(y = y_i | \mathbf{v}_I)$. Alternatively, the prediction can be represented as an embedding vector $\mathbf{v}_e$ of composition of one or several text embeddings. Prediction in terms of $\mathbf{v}_e$ enables the exemplar model to support zero-shot prediction for labels similar to exemplar labels.

**KNN**: Given $\mathbf{v}_I$, the KNN memory module finds its most similar $k$ entries in the memory through cosine similarities between $\mathbf{v}_I$ and all $\mathbf{v}_{I_j}$ in the memory. Let $\mathcal{N}_k(\mathbf{v}_I)$ be the set of indices of the $k$ highest cosine similarity scores to $\mathbf{v}_I$. KNN classification for $\mathbf{v}_I$ can be performed by majority voting from the values: $\hat{y} = \arg\max_y \sum_{j \in \mathcal{N}_k(\mathbf{v}_I)} \mathbb{1}(y_j = y)$. Here, $\mathbb{1}(\cdot)$ is an indicator function. The probability of $\mathbf{v}_I$ being label $y_i$ is $p_e(y = y_i | \mathbf{v}_I) = \frac{1}{k} \sum_{j \in \mathcal{N}_k(\mathbf{v}_I)} \mathbb{1}(y_j = y_i)$.

To predict the embedding $\mathbf{v}_e$, we can use the text embedding of the most likely label. But we find that computing a similarity-weighted average of the retrieved text embeddings gives better performance: $\mathbf{v}_e = \sum_{j \in \mathcal{N}_k(\mathbf{v}_I)} \beta_j \cdot f_{\text{txt}}(t_j)$, where $\beta_j \propto \exp(l(\mathbf{v}_I, \mathbf{v}_{I_j}))$ and $\mathbf{v}_{I_j}$ is the image embedding the $j$th neighbor.

KNN takes virtually no time to train ($\mathcal{O}(1)$), and reasonably fast retrieval is possible with optimized libraries and parallel computing. However, accuracy tends to be lower than model-based methods.

**Linear probe**: Parameterized models offer an alternative approach, learning a fixed set of parameters that exploit the underlying structures of the data to maximize classification performance. The linear probe (LinProbe) method runs on extracted image embeddings and trains linear classifiers on all accumulated exemplars when new exemplars are added. The output probability can be written as: $p_e(y = y_i|\mathbf{v}_I; \boldsymbol{\theta}) = \frac{\exp(\boldsymbol{\theta}_{y_i}^T \mathbf{v}_I)}{\sum_{y_j \in \mathbf{Y_e}} \exp(\boldsymbol{\theta}_{y_j}^T \mathbf{v}_I)}$, where $\boldsymbol{\theta}_{y_i}$ represents the learned model parameters for label $y_i$ and $\mathbf{Y}_e$ is the union of exemplar labels. The predicted embedding is the text embedding of the label $\hat{y}$ with maximum probability: $\mathbf{v}_e = f_{\text{txt}}(t_{\hat{y}})$.

Compared to KNN, the LinProbe model is much slower to train — $\mathcal{O}(n)$ for $n$ training samples assuming a constant number of epochs, which may be prohibitive when the model needs to be updated quickly based on few examples. However, classification accuracy tends to be higher.

**Tree probe**: In a continual learning setting, we would ideally have fast training time of KNN with the relatively good accuracy of LinProbe. We take inspiration from the instance-based and lazy learning literature (Aggarwal, 2014), particularly locally linear models (Domeniconi & Gunopulos, 2002; Atkeson et al., 1997). These methods classify a test sample by finding its $k$-nearest neighbors and applying a linear classifier trained on the neighbors. This achieves $\mathcal{O}(1)$ training time but may be impractical for inference, since a new classifier may need to be trained for each test sample.

Instead, we propose an approximation, building a clustering tree from the training data and training a linear classifier in each leaf node, as shown in Fig. 2. Starting from a root node, we search for the nearest leaf node for a new data point and insert it if the node has not reached the predefined node capacity $\psi$. If a leaf node reaches $\psi$, it splits into two child nodes and becomes a non-leaf node. The attached data points are distributed to their children by KMeans clustering. In experiments, when receiving new data, samples are added into the cluster tree one by one. Only classifiers in affected leaf nodes need to be retrained. When fixing the number of linear model training epochs and KMeans iterations, the complexity to incorporate a new exemplar in training is $\mathcal{O}(\psi + \log n)$ with $\psi >> \log n$; the training time stays limited even when the total number of exemplars is very large.

The simplest inference method would be to assign a test sample to a leaf node in the cluster tree and classify it within the corresponding linear model, but this may lead to non-smooth predictions for samples near the cluster boundaries. In our experiments, we ensemble the classifiers from leaf nodes corresponding to the $k$ nearest neighbors, such that the final output probability $p_e(y = y_i|\mathbf{v}_I)$ is the average of the probabilities predicted by each neighbor's classifier.

Similar to KNN, the exemplar embedding $\mathbf{v}_e$ can be computed as the text embedding of the most likely label from all neighbor classifiers. However, we obtain the best performance using a similarity-weighted average embedding from the most likely label $\hat{y}_j$ of each classifier in the retrieval set: $\mathbf{v}_e = \sum_j \beta_j \cdot f_{\text{txt}}(t_{\hat{y}_j})$, where $\beta_j$ is similarly defined as in the KNN classification. The tree-probe method, denoted TreeProbe, achieves similar accuracy to LinProbe in our continual learning experiments with sufficiently large $\psi$, but with much faster training time.

### 3.3 Fusing predictions of the two systems

We want to integrate predictions from the zero-shot and exemplar-based models to retain good open-vocabulary classification performance and improve accuracy in predicting exemplar labels. This can be tricky, especially when some labels in the task are not contained within the exemplar labels.

One approach is to simply average the probability or embedding predictions of the two models:

$$p(y = y_i|I) = \alpha p_e(y = y_i|\mathbf{v}_I) + (1 - \alpha)p_z(y = y_i|I) \tag{3}$$

$$\mathbf{v}_{\text{out}} = \alpha \mathbf{v}_e + (1 - \alpha)\mathbf{v}_I \tag{4}$$

When setting $\alpha = 0.5$ for an unweighted average, this approach is denoted as `Avg-Prob` or `Avg-Emb`, respectively. Using an unweighted average presumes that the zero-shot and exemplar models are equally reliable for all samples. However, if the test sample's label is within the exemplar's domain, the exemplar model will tend to

be more accurate; otherwise, the zero-shot model will almost certainly outperform. The test label is unknown, but an educated guess could enable a better prediction.

Addressing this issue, we devise an adaptive weighting mechanism, named **A**daptive **I**nstance **M**arginalization (**AIM**), that estimates the likelihood of a test sample's label being in the exemplar set and balances the predictions from both models accordingly. The target label set is divided into exemplar $y \in \mathbf{Y}_e$ and non-exemplar $y \notin \mathbf{Y}_e$ subsets, where $\mathbf{Y}_e$ is the union of all exemplar labels.

The likelihoods $p(y \in \mathbf{Y}_e|I)$ and $p(y \notin \mathbf{Y}_e|I)$ are obtained by summing the zero-shot probabilities over these subsets: $p(y \in \mathbf{Y}_e|I) = \sum_{i \in \mathbf{Y}_e} p_z(y = y_i|I)$ and $p(y \notin \mathbf{Y}_e|I) = \sum_{i \notin \mathbf{Y}_e} p_z(y = y_i|I) = 1 - p(y_i \in \mathbf{Y}_e|I)$, with the summation in the denominator of Eq. 2 over all candidate labels for the current task. Naturally, we set $\alpha = p(y \in \mathbf{Y}_e|I)$ in Eq. 4, which capitalizes on the strengths of both models, improving overall performance.

Our final `AIM-Prob` method further encodes that both zero-shot and exemplar models are predictive for exemplar classes, while only the zero-shot model can be trusted for non-exemplar classes:

$$p(y = y_i|I) = p(y \in \mathbf{Y}_e|I) \times \frac{p_z(y = y_i|I)p_e(y = y_i|\mathbf{v}_I)}{\sum_{j \in \mathbf{Y}_e} p_z(y = y_j|I)p_e(y = y_j|\mathbf{v}_I)} + p(y \notin \mathbf{Y}_e|I) \times p_z(y = y_i|I). \tag{5}$$

## 4 Experimental setup

### 4.1 Tasks

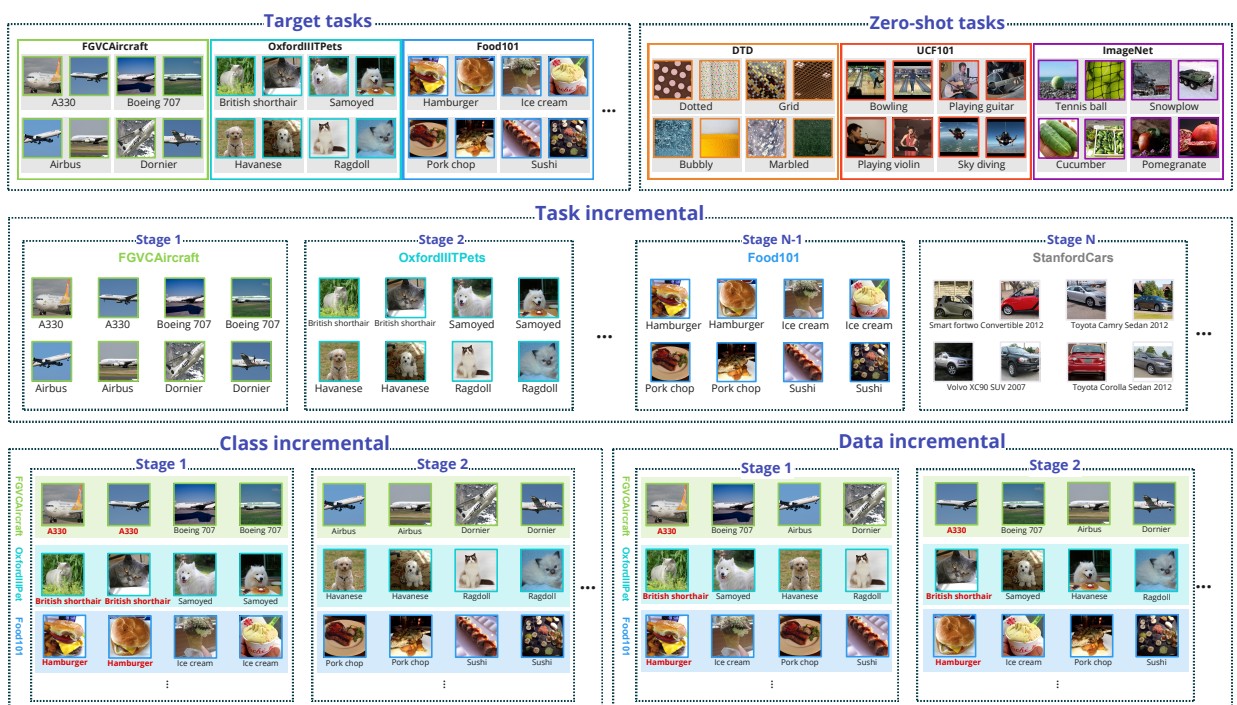

Figure 3: The **upper** row illustrates several randomly selected samples in target tasks and zero-shot tasks. In the **middle** and the **lower** row, we illustrate how data samples are organized in task, class and data incremental learning scenarios. Borders of images with different colors indicates the source of the data. Red class names in data and class incremental learning are used to highlight differences of the two settings.

We evaluate our system on target and zero-shot classification tasks. A "task" is an assignment of an image into a label from particular label set. The exemplar model has received examples in the context of "target" tasks but not "zero-shot" tasks. We utilize general tasks such as ImageNet (Russakovsky et al., 2015), SUN397 (Xiao et al., 2010), CIFAR100 (Krizhevsky & Hinton, 2009), and fine-grained tasks like EuroSAT (Helber et al.,

2019), OxfordIIITPets (Parkhi et al., 2012), DTD (Cimpoi et al., 2014), Flower102 (Nilsback & Zisserman, 2008), FGVCAircraft (Maji et al., 2013), StanfordCars (Krause et al., 2013), Food101 (Bossard et al., 2014), UCF101 (Soomro et al., 2012).

We report main results using **target tasks** CIFAR100, SUN397, FGVCAircraft, EuroSAT, OxfordIIITPets, StanfordCars, Food101 and Flowers102 and **zero-shot tasks** ImageNet, UCF101, and DTD. All target tasks collectively provide > 220k samples in total, which is sufficient to give a reliable assessment of different methods. Several randomly selected samples of some tasks are displayed at Fig. 3. The supplemental material contains dataset descriptions, prompt templates, zero-shot performance of tasks, and additional experimental results and analysis.

## 4.2 Evaluation scenarios

We consider several continual learning scenarios for receiving data: 1) **Data incremental**: A fraction of the training data, randomly sampled without enforcing class balance, is added in each stage; 2) **Class incremental**: All training data for a randomly sampled subset of classes are added in each stage; 3) **Task incremental**: All data for a single task, i.e. a dataset of examples assigned to a set of target labels, are added in each stage. Data incremental learning includes seven stages, comprising 2%, 4%, 8%, 16%, 32%, 64%, and 100% of task data, respectively. Class incremental learning divides a task into five stages, each containing 20% of classes. In task incremental learning, each task is considered a stage. In data and class incremental experiments, models are built separately for each target task. A target task is fully evaluated if there is at least one training sample for that task, even if there are no training samples for some classes. In task incremental, one model is built spanning all accumulated labels in each stage. In all cases, results are reported as the average accuracy of target and zero-shot tasks at each stage. We additionally compute the averaged accuracy on seen classes and unseen classes of each target task for class incremental learning to give more detailed result analysis. An intuitive demonstration of these continual learning scenarios are shown in Fig. 3.

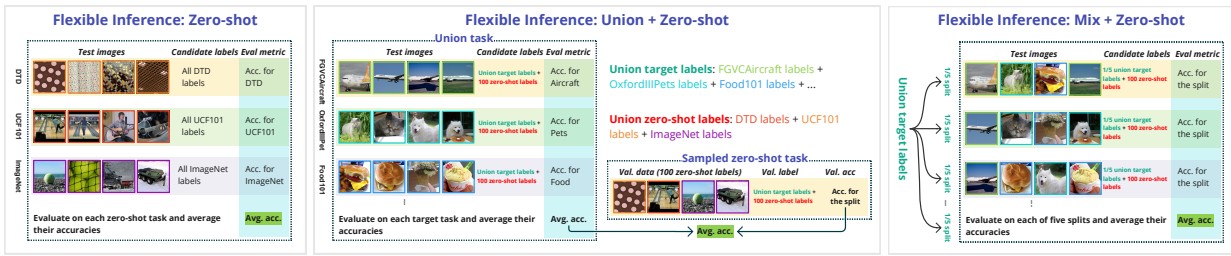

Figure 4: Demonstration of the three inference scenarios of flexible inference. Images and visual elements are consistent with Fig. 3. Bold text in green shows the final accuracy obtained from each scenario.

**Flexible inference:** In the task incremental setting, after all training data for target tasks is received, we will evaluate each method's performance in three inference scenarios:

- **Zero-shot:** evaluate on each zero-shot task and then average the accuracy on all tasks. Under this scenario, higher performance of a model than CLIP would indicate positive transfer from learned tasks to zero-shot tasks, while lower performance would suggest forgetting has occurred.
- **Union + Zero-shot:** we first create a union of target task labels that is considered as candidate labels for each target task. Then, we randomly sample 100 labels from the union of zero-shot task labels. We separately compute the average accuracies on the target and zero-shot tasks. The final performance is the average of the target and zero-shot accuracy.
- **Mix + Zero-shot:** we randomly split the union of target labels into five splits. For each split, we additionally add a random sample of 100 labels from the zero-shot task. To balance evaluation across classes, we draw 100 test samples per class. Performance is the average across all splits.

Fig. 4 gives a more intuitive demonstration of these three inference scenarios. These scenarios are especially challenging for most continual learning methods because there is no task identifier at inference time, and the label of a test sample will sometimes be among the trained classes and sometimes not.

## 5 Experimental results

### 5.1 Effectiveness of complementary system and AIM

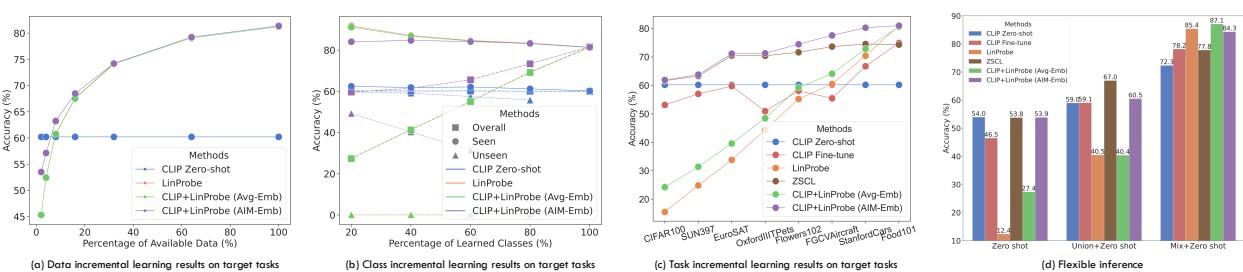

Figure 5: **(a)** Results comparing CLIP zero-shot, LinProbe and CLIP+LinProbe with Avg-Emb and AIM-Emb on target tasks under the data incremental learning scenario. **(b)** Results of corresponding models on target tasks under class incremental learning. We visualize curves for seen and unseen classes including the overall performance identified with different markers. **(c)** Results of corresponding models on target tasks under task incremental learning, along with the performance of fine-tuning the whole CLIP network (CLIP Fine-tune) and ZSCL (Zheng et al., 2023). **(d)** Flexible inference results after task incremental learning on all tasks. Note LinProbe is hard to see in (a) and (b) because it has similar results as CLIP+LinProbe (Avg-Emb).

We first investigate the importance of complementary learning systems and the effectiveness of AIM. We design experiments comparing CLIP Zero-shot model, exemplar model using LinProbe and then complementary models that incorporate CLIP Zero-shot and LinProbe using the Avg-Emb and AIM-Emb fusing operations, denoted by CLIP+LinProb (Avg-Emb) and CLIP+LinProbe (AIM-Emb).

The results on target tasks under data, class, and task incremental learning scenario are shown in Fig. 5 (a-c). For data incremental, except for CLIP zero-shot, all other methods constantly improve on target tasks, with CLIP+LinProbe (AIM-Emb) being the best on all stages. When the percentage of available data is larger than 20%, all other models surpass the zero-shot model. LinProbe and CLIP+LinProbe (Avg-Emb) have similar performances on target tasks, which can also be observed in the class incremental learning results in Fig. 5 (b).

Results under class incremental learning are crucial as this setup simulates scenarios where exemplar labels do not encompass all candidate labels. In such cases, we cannot simply decide between using the exemplar model or zero-shot model for prediction by checking if exemplar labels fully overlap with candidate labels. In Fig. 5 (b), we plot the accuracies on seen and unseen classes along with the overall accuracy on all classes. The zero-shot model underperforms other methods on seen classes but excels in unseen classes. LinProbe achieves high accuracy on seen classes but performs poorly on unseen ones, with accuracy close to 0. CLIP+LinProbe (Avg-Emb) shows similar results. The AIM-Emb version has slightly lower accuracy than the Avg-Emb version on seen classes in early stages (<60% classes) but maintains reasonable performance on unseen classes, particularly in earlier stages. Its unseen accuracy decreases as more classes are learned because AIM gradually favors the exemplar model. Overall, CLIP+LinProbe (AIM-Emb) demonstrates the highest overall accuracy among all methods, highlighting the effectiveness of AIM.

In Fig. 5 (c) & (d) under the task-incremental learning and flexible inference scenario, in addition to the models mentioned above, we also compare with ZSCL (Zheng et al., 2023). For the implementation, we try to maintain identical training techniques and hyperparameters as in the original codes of ZSCL, and we will release codes for sanity check with written details in the supplemental material. Furthermore, we train CLIP by fine-tuning all its parameters except for the logit scale on target datasets, resulting in a stronger exemplar

model—CLIP Fine-tune. As a side-to-side comparison with ZSCL, we only use the data of the current stage to train this model but without additional reference datasets used in ZSCL.

From the results in Fig. 5 (c), the AIM approach steadily shows better performance than the zero-shot model at all stages. CLIP Fine-tune, LinProbe and CLIP+LinProbe (Avg-Emb) can only beat the zero-shot model after the 6-th stage (FGVCAircraft). However, CLIP Fine-tune is better than LinProbe and CLIP+LinProbe (Avg-Emb) in early stages, implying that the loss of generalization to other tasks is less severe. ZSCL achieves relatively close performance as CLIP+LinProbe (AIM-Emb) in early stages but lags behind quickly in latter ($\sim -6\%$ off in the final stage). Overall, CLIP+LinProbe (AIM-Emb) has the best performance across all stages and show better performance than exemplar-only models, demonstrating the benefits of complementary systems.

As shown in Fig. 5 (d), exemplar-only models, being biased towards learned classes, are generally incompetent on zero-shot tasks. LinProbe is extremely bad on zero-shot tasks, leading to poor performance on both Zero-shot and Union+Zero-shot scenarios where zero-shot tasks are evaluated. However, CLIP+LinProbe (Avg-Emb) shows considerable improvement over LinProbe in Zero-shot and Mix+Zero-shot scenarios while remaining comparable in Union+Zero-shot. CLIP Fine-tune received less severe impact of biased learning to target classes but still has a decent decrease under Zero shot. When replacing the fusing operation with AIM-Emb, performances across all scenarios significantly improve. Note that ZSCL also has good Zero-shot performance, as mentioned in its paper, and obtains the best Union+Zero-shot performance in our evaluation. Our method is inferior to ZSCL under the Union+Zero-sho scenario mainly because AIM likely assigns more weights to the exemplar model due to the large portion of in-exemplar labels (around 91%) of all candidate labels. Overall, our system shows a decent flexible inference ability with the effectiveness of the complementary systems well validated.

## 5.2 Comparison of Efficiency and Accuracy for exemplar-based Approaches

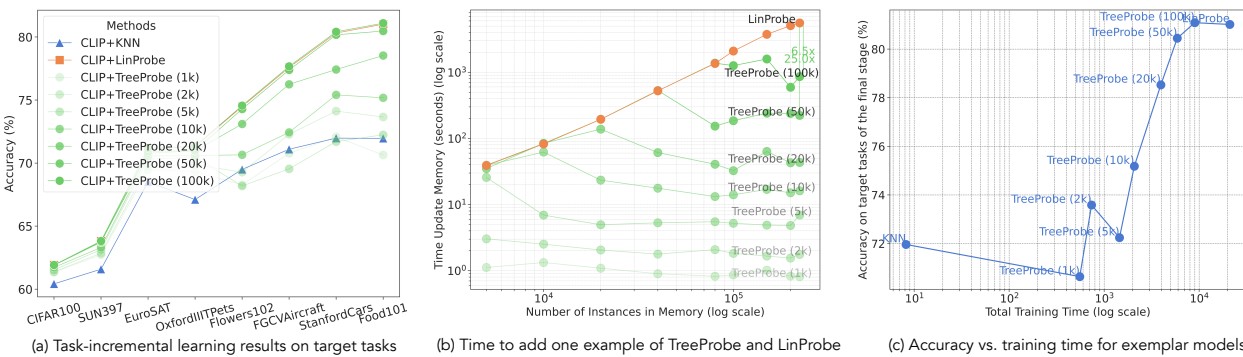

(a) Task-incremental learning results on target tasks

(b) Time to add one example of TreeProbe and LinProbe

(c) Accuracy vs. training time for exemplar models

Figure 6: **(a)** Results of different complementary models under the task-incremental learning scenario. Numbers included in a bracket are the node capacity of corresponding TreeProbe models. **(b)** We perform an efficiency analysis of TreeProbe with different node capacities and LinProbe by estimating how much time needed to incorporate one example into the exemplar model with different number of instances carried in the exemplar set. To minimize randomness, each method is run five times. At each data collection point, we sample an example from the exemplar set and fit it to memory five times. Thus, each data point averages 25 unique simulations. $x$ and $y$-axes are visualized in log scale. **(c)** The accuracies of different complementary models on target tasks after learning the final stage ($y$-axis) *vs.* the total training time in log scale ($x$-axis).

In this part, we perform a thorough analysis of different complementary models using AIM on their performances and efficiency: CLIP+KNN, CLIP+LinProbe and CLIP+TreeProbe. We can also vary the node capacities of TreeProbe to create different versions of CLIP+TreeProbe models.

We compare their performance under the task incremental learning scenario. Results in Fig. 6 (a) show that node capacity can be used to control the performance of CLIP+TreeProbe. When node capacity exceeds 5k, TreeProbe steadily outperforms KNN, and when it reaches 100k, TreeProbe slightly surpasses LinProbe.

Fig. 6 (b) illustrates that LinProbe adheres to linear learning complexity as the number of instances in the exemplar set increases. When node capacity is not reached, TreeProbe only has one classifier, thus identical to LinProbe in learning complexity. After reaching node capacity, TreeProbe exhibits constant learning complexity. TreeProbe (100k) and (50k) achieve similar performance to LinProbe but with 6x and 25x improvement in learning speed. This corroborates our analysis of the time complexity of learning for different exemplar models in Sec. 3.2. Fig. 6 (c) combines the results of Fig. 6 (a) and Fig. 6 (b), showing the trade-off of learning efficiency and accuracy. The TreeProbe method, therefore, offers flexibility in selecting the node capacity that appropriately balances responsiveness with performance for a specific application.

We also compare the learning efficiency of our methods with CLIP Fine-tune and ZSCL (Zheng et al., 2023) and summarize the training time in Tab. 1. Since ZSCL and CLIP Fine-tune require GPU accelerated training, we also implement GPU-versions of TreeProbe and LinProbe for better comparison. Since our methods only operate on the image embeddings, we can cache the embeddings for new samples and store for expedited replay or multi-epoch training. For both our models, we train the linear classifiers for 20 epochs, using hyperparameters selected from a hyperparameter sweep. The evaluation is taken using the task incremental learning scenario and the training time is accumulated for all eight target tasks. From the table, after taking the processing time to get image embeddings in account, our approaches are still much faster than CLIP Fine-tune and ZSCL. Note that to mitigate forgetting, ZSCL needs to refer to the outputs of frozen CLIP to calculate distillation losses, making it around two times slower than CLIP Fine-tune. Besides training efficiency, our approaches are more effective through the final averaged accuracy over all target tasks. This test further reinforces the training efficiency and effectiveness of TreeProbe.

| Method | Training Time (s) | △ | Avg. Acc. (%) | △ |
|--------|------------------|-----|---------------|------|
| TreeProbe (50k) | 239.1 | 0.0 | 80.2 | 0.0 |
| LinProbe | 256.6 | +17.5 | 80.4 | +0.2 |
| CLIP Fine-tune | 2542.7 | ×10.6 | 74.9 | -5.3 |
| ZSCL | 5409.3 | ×22.6 | 74.3 | -5.9 |

Table 1: Efficiency comparison of different methods. Training Time is measured in seconds, and Avg. Acc. is the averaged accuracy over all target tasks evaluated at the end of task-incremental learning. △ shows the difference from the baseline method TreeProbe (50k) ("×" means times). For equal comparison using the same GPU, the linear layers in TreeProbe and LinProbe are implemented using PyTorch's linear layer (i.e, `torch.nn.Linear`) to support GPU accelerated training.

## 5.3 Comparison to previous methods

| Method | Transfer | △ | Avg. | △ | Last | △ |
|--------|----------|------|------|-------|------|-------|
| CLIP Zero-shot | 69.4 | 0.0 | 65.3 | 0.0 | 65.3 | 0.0 |
| LwF (Li & Hoiem, 2016) | 56.9 | -12.5 | 64.7 | -0.6 | 74.6 | +9.3 |
| iCaRL (Rebuffi et al., 2017) | 50.4 | -19.0 | 65.7 | +0.4 | 80.1 | +14.8 |
| WiSE-FT (Wortsman et al., 2022a) | 52.3 | -17.1 | 60.7 | -4.6 | 77.7 | +12.4 |
| ZSCL (Zheng et al., 2023) | 68.1 | -1.3 | 75.4 | +10.1 | 83.6 | +18.3 |
| TreeProbe (50k) | **69.3** | **-0.1** | **75.9** | **+10.6** | **85.5** | **+20.2** |

Table 2: Comparison of different methods on MTIL in Order I from ZSCL (Zheng et al., 2023). All results are taken from other ZSCL's paper except for TreeProbe. "Transfer" evaluates the model's performance on zero-shot tasks; "Last" is the averaged accuracy on all target tasks after finishing the final task while "Avg." computes the average task performance on all training stages.

As introduced in Sec. 2.2, ZSCL (Zheng et al., 2023) explores a similar setting to open-vocabulary continual learning and is mainly evaluated under the task-incremental learning scenario. To ensure a fair comparison, we evaluate our approach using their evaluation protocols, employing the same prompt ensemble technique, data split ratio and seed, backbone network (ViT-B/16), and other relevant parameters. The results are

presented in Tab. 2. Our TreeProbe (50k) model achieves the best performance across all metrics. Compared to the CLIP zero-shot model, our method barely compromises on Transfer, demonstrating robust performance on zero-shot tasks. Notably, our method is more lightweight in GPU consumption than ZSCL. Additionally, we do not perform hyperparameter selection for each task, as we assume no prior knowledge of future tasks in practice, whereas ZSCL used different learning rates for different tasks to optimize performance.

### 5.4 Scaling to larger zero-shot models

| Method | ViT-B/32 | | ViT-L/14@336px | | ViT-H/14 | |
|---|---|---|---|---|---|---|
| | Target | Zero-shot | Target | Zero-shot | Target | Zero-shot |
| CLIP Zero-shot | 60.2 | 54.0 | 72.1 | 65.6 | 79.2 | 71.7 |
| TreeProbe (50k) | 80.5 | 53.8 | 86.6 | 65.4 | 90.2 | 71.6 |

Table 3: Accuracies of CLIP Zero-shot and TreeProbe on target and zero-shot tasks with pretrained image encoders of different capacities.

We use the CLIP ViT/B-32 model as our default zero-shot model but can easily adapt to other models. We experiment with the pre-trained CLIP model ViT-L/14@336px, which boasts approximately 4x the capacity of ViT-B/32, and ViT-H/14, which is double the size of ViT-L/14@336px. We present our findings in Tab. 3, where both methods are evaluated under the task incremental learning scenario. Performance improves for larger models, and TreeProbe consistently outperforms CLIP Zero-shot on the target tasks with nearly identical performance on zero-shot tasks. This demonstrates that our approach is beneficial across a wide range of zero-shot model sizes.

## 6 Conclusion and limitations

In this work, we present an efficient and performant "tree probe" exemplar-based model and Adaptive Instance Marginalization to combine zero-shot and exemplar models for open-vocabulary continual learning. Our method is able to efficiently incorporate new samples, improving performance for related tasks without negatively impacting zero-shot task performance, providing a simple way to continually improve large-scale pretrained image-text classification models. We believe this work is a step towards more flexible, efficient, and effective strategies in open-vocabulary continual learning.

Our work has several **limitations**. First, we do not consider constraints in how many exemplars can be stored. Since each exemplar requires storing up to 4KB for the image and text encodings (using the base CLIP ViT-B/32 model without any compression or limiting precision), roughly one million exemplars can be stored per 4GB of memory. Second, we do not investigate how to improve the consolidated zero-shot model. This is a promising direction for future work. Finally, we do not explore structured prediction problems like semantic segmentation or visual question answering, which likely require more complex image representations than a single embedding vector.

### Acknowledgments

This work is supported in part by ONR award N00014-21-1-2705, ONR award N00014-23-1-2383, and U.S. DARPA ECOLE Program No. #HR00112390060. The views and conclusions contained herein are those of the authors and should not be interpreted as necessarily representing the official policies, either expressed or implied, of DARPA, ONR, or the U.S. Government. The U.S. Government is authorized to reproduce and distribute reprints for governmental purposes notwithstanding any copyright annotation therein.

### Broader Impact Statement

This paper presents an efficient and flexible continual learning system for open-vocabulary image classification. Our method may reduce the computational cost of extending and improving large foundation models, which could reduce the carbon footprint of development and facilitate new applications. As with other image

classification approaches, our method is dependent on training data and annotations, which may be biased or non-representative of deployed use cases.

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

## A   Appendix

## S-1   Algorithmic descriptions of TreeProbe

---

**Algorithm 1** Training Procedure of TreeProbe

---

**Require:** Training set $X$, Tree $T$, Leaf capacity $\psi$
**Ensure:** Trained classifiers in each leaf node of $T$
 1: **for all** $\mathbf{v}_i \in X$ **do**
 2:     $l = \text{NEARESTLEAF}(\mathbf{v}_i, T)$
 3:     **if** $\text{COUNT}(l) < \psi$ **then**
 4:         $l = \text{INSERTDATA}(\mathbf{v}_i, l)$
 5:         $\text{TRAINCLASSIFIER}(l)$
 6:     **else**
 7:         $\text{SPLITNODE}(l, \mathbf{v}_i)$
 8:         $l = \text{NEARESTLEAF}(\mathbf{v}_i, T)$
 9:         $l = \text{INSERTDATA}(\mathbf{v}_i, l)$
10:         $\text{TRAINCLASSIFIER}(l)$
11:     **end if**
12: **end for**

---

**Algorithm 2** Inference Procedure of TreeProbe

---

**Require:** Image embedding $\mathbf{v}_I$, Tree $T$, Number of nearest nodes $k$, Exemplar set $M$
**Ensure:** Exemplar embedding $\mathbf{v}_e$ for $\mathbf{v}_I$
 1: $\mathbf{v}_{\mathcal{K}} = \text{FINDNEARESTSAMPLES}(\mathbf{v}_I, M)$
 2: $\mathbf{v} = \text{list}()$
 3: **for all** $\mathbf{v}_i \in \mathbf{v}_{\mathcal{K}}$ **do**
 4:     $l = \text{NEARESTLEAF}(\mathbf{v}_i, T)$
 5:     $c = \text{GETCLASSIFIER}(l)$
 6:     $\mathbf{v} \leftarrow \text{CLASSIFY}(\mathbf{v}_I, c)$
 7: **end for**
 8: $\mathbf{v}_e = \text{COMPUTEEMBEDDING}(\mathbf{v}, \mathbf{v}_I)$

---

Algorithm 1 and Algorithm 2 contain psuedocode for the training and inference of our TreeProbe method. Definitions of the involved functions are provided below:

- **NearestLeaf**($\mathbf{v}_i$, $T$): Returns the nearest leaf node to the data point $x_i$ in tree $T$.

- **Count**($l$): Returns the current number of data points in leaf node $l$.

- **InsertData**($\mathbf{v}_i$, $l$): Inserts data point $\mathbf{v}_i$ into leaf node $l$ and returns the updated node.

- **SplitNode**($l$, $\mathbf{v}_i$): Splits leaf node $l$ into two child nodes when it reaches capacity, distributes data points using KMeans clustering, and adds new data point $\mathbf{v}_i$ to the appropriate child node.

- **TrainClassifier**($l$): Trains a linear classifier on the data points in leaf node $l$.

- **FindNearestSamples**($\mathbf{v}_I$, $M$): Finds the $k$ nearest neighbors in the exemplar set to the image embedding $\mathbf{v}_I$.

- **GetClassifier**($l$): Return the classifier for node $l$.

- **Classify**($\mathbf{v}_i$, $c$): Classifies $\mathbf{v}_i$ using the linear classifier $c$, returning the label embedding of the most likely label.

- **ComputeEmbedding**($\mathbf{v}$, $\mathbf{v}_I$): Computes the exemplar embedding for $\mathbf{v}_I$ by applying a similarity-weighted average of the text embeddings of the most likely class labels from a temporal list $\mathbf{v}$.

The notation of Algorithm 1 and Algorithm 2 may differ from the main paper.

## S-2 Implementation details

We conduct our experiments on a setup featuring an RTX 3090 GPU and an AMD Ryzen 9 5950X CPU, using PyTorch as our primary framework. We adhere to the CLIP code example, using sklearn `LogisticRegression` to implement linear classifiers and setting the sklearn regularization strength to 0.316. The maximum iteration is set to 5k. Our tree probe's node capacity is set at 50k. For efficient retrieval from large-scale exemplar sets, we use FAISS (Johnson et al., 2019), specifically using the IndexFlatIP class for its precision and performance. Model performances are gauged via Top-1 accuracy, with the officially released ViT-B/32 CLIP checkpoint serving as our memory or zero-shot model. We select $k = 9$ based on a hyperparameter sweep. Our approach is not sensitive to $k$, with very similar performance in a range from 6 to 30.

**Additional implementation details of CLIP Fine-tune and ZSCL.** Since we do not assume to have additional knowledge about data distributions, we choose the learning rate for both methods as the most frequent used learning rate in ZSCL experiments, i.e., 1e-5. For each task, following ZSCL, we warmup training for 100 iterations and proceed to train 1K iterations. For the weight ensemble technique used in ZSCL, we also use an update interval of 100 iterations. Batch size is kept to 64 for both methods, with AdamW (Loshchilov & Hutter, 2019) optimizer and the beta set to 0.9. The base network backbone we use for both methods is CLIP ViT-B/32, identical to our approach.

**Additional implementation details of GPU-version TreeProbe/LinProbe.** After sweeping a good set of hyperparameters that lead to smaller number of training epochs while maintaining good performance on held-out classification datasets (Caltech101), we set learning rate to 0.005, optimizer to SGD, weight decay to 0.1, and use 20 epochs for training each stage. We use the cross-entropy loss for training the probes.

## S-3 More experiment setting details

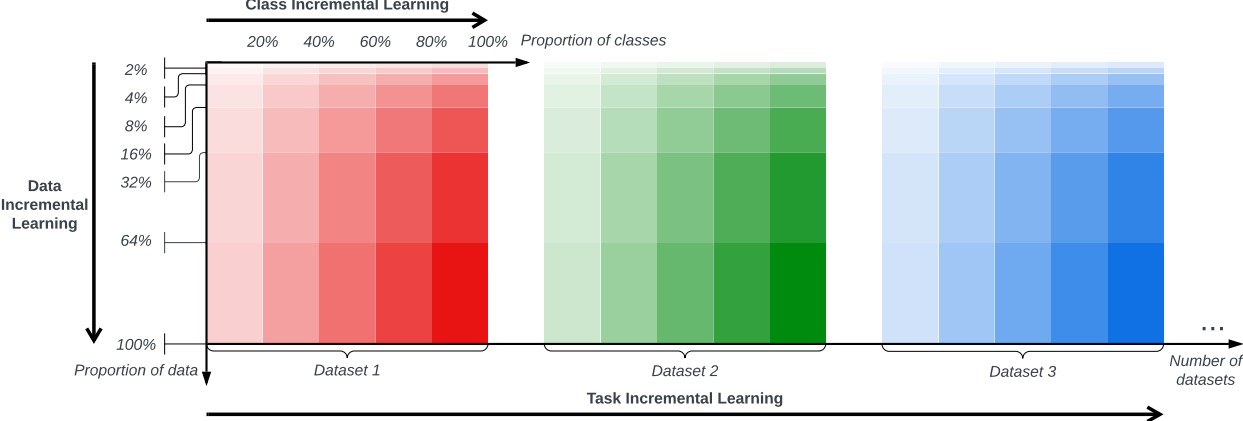

Figure S1: Illustration of continual learning scenarios. Data incremental learning includes seven stages, each comprising 2%, 4%, 8%, 16%, 32%, 64%, and 100% of task data respectively. Class incremental learning divides a task into five stages, each containing 20% of classes. In task incremental learning, each task is considered a stage.

We present an illustration of data, class, and task incremental learning scenarios in Fig. S1. When evaluating different methods in data and class incremental learning scenarios, we ensure fairness by randomly selecting an identical portion of data/class for all methods, achieved by setting the same seed. Models are separately built for each target task. The performance of each stage is averaged across all target tasks. In task-incremental

learning, each stage is embodied by a distinct task. The task order is randomly arranged as CIFAR100, SUN397, FGVCAircraft, EuroSAT, OxfordIIITPets, StanfordCars, Food101, and Flowers102, consistent for all methods for fair comparison. As shown in ZSCL (Zheng et al., 2023), task order has little impact on the relative performance comparison of different models. For all learning scenarios, we make the assumption that training data accumulates across all stages in all scenarios, with a similar spirit to (Prabhu et al., 2023). This assumption is based on the fact that real-world applications are often more limited by computational and time budgets than by storage. If data privacy is not a concern, building a continual learning system without losing data access is more effective than assuming past data is non-accessible. Furthermore, we enhance storage efficiency by saving samples as condensed feature vectors, a significant improvement over some earlier works.

## S-4    Descriptions of tasks

We perform experiments on various commonly used visual datasets to demonstrate the generalization capabilities of our method. These datasets encompass a broad range of image categories and application scenarios, including both fine-grained and generalized datasets. We briefly introduce all used tasks in this paper in the following.

### S-4.1    General tasks

**ImageNet** ImageNet (Russakovsky et al., 2015) contains 1,281,167 training images, 50,000 validation images and 100,000 test images. The categories represent a wide variety of objects, animals, scenes, and even abstract concepts. This dataset has served as a fundamental dataset to evaluate performances of classification models, or as a pretraining dataset.

**CIFAR100** The CIFAR100 dataset (Krizhevsky & Hinton, 2009) consists of object images and is a subset of the 80 million tiny images dataset. It contains 60,000 $32 \times 32$ color images from 100 object categories, with 600 images per category. The dataset has 100 fine-grained classes, grouped into 20 coarse-grained classes.

**SUN397** The SUN397 dataset (Xiao et al., 2010) consists of scene images, containing 108,754 images across 397 scene categories, with each category having between 100 and 500 images. This dataset is commonly used for scene understanding tasks. Since there is no official dataset split for this dataset, we randomly select 60% of images as training data, 20% as validation data, and the rest as test data. We use NumPy random permutation to split with the seed set to 0.

### S-4.2    Fine-grained tasks

**FGVCAircraft** The FGVCAircraft dataset (Maji et al., 2013) serves as a benchmark for fine-grained visual categorization of aircraft. It contains 10,200 images from 102 distinct categories. Each category includes approximately 100 images, annotated with the aircraft model, variant, and manufacturer.

**DTD** The Describable Textures Dataset (DTD) (Cimpoi et al., 2014) consists of 5,640 images across 47 texture categories, with each category featuring 120 real-world texture images such as fabrics, rocks, and surfaces. The dataset poses a challenge for texture classification due to subtle differences between textures within the same category and large variations in texture appearance caused by scale, orientation, and lighting.

**Food101** The Food-101 dataset (Bossard et al., 2014) comprises 101,000 images across 101 food categories, each with 1,000 images. This dataset challenges fine-grained image classification due to high intra-class variation and visual similarities across categories. It serves as a rigorous benchmark for evaluating computer vision models in food recognition and provides a robust platform for training machine learning models in understanding culinary aesthetics and preferences.

**StanfordCars** The StanfordCars dataset (Krause et al., 2013) is a benchmark dataset containing 16,185 images from 196 different car classes, divided into a 50-50 training and testing split. The classes correspond to specific car makes, models, and years, such as the 2012 Tesla Model S or 2012 BMW M3 coupe.

**Flowers102** The 102 Category Flower Dataset (Nilsback & Zisserman, 2008) is a compilation of flower images. It includes 8,189 images across 102 flower categories, with each category containing between 40 and 258 images. The dataset's images vary in size and aspect ratio, captured using different cameras, lighting conditions, and backgrounds.

**OxfordIIITPets** The OxfordIIITPets dataset (Parkhi et al., 2012) is a collection of pet images, featuring 7,349 images from 37 different cat and dog breeds. Each breed has between 100 and 200 images. The dataset is challenging because the appearance of the same breed can vary significantly, and different breeds may have similar-looking features.

**EuroSAT** The EuroSAT dataset (Helber et al., 2019) is a remote sensing image dataset comprising Sentinel-2 satellite data. It contains 27,000 images that cover 13 spectral bands and consist of 10 different land use and land cover categories, including forests, urban areas, and water bodies. This dataset is commonly employed for remote sensing and land cover classification tasks. Since there is no official dataset split for this dataset, we randomly select 70% of images as training data and the rest as validation data. We use NumPy random permutation to perform splitting with the seed set to 0.

**UCF101** The UCF101 dataset (Soomro et al., 2012) is a commonly used benchmark for action recognition. It consists of 13,320 videos from 101 action categories, with each category containing at least 100 videos. The actions include a wide range of human activities such as basketball shooting, horse riding, and juggling. The dataset is unique in its focus on complex, naturalistic action sequences, with videos varying in length from a few seconds to a minute. Since there is no official dataset split for this dataset, we randomly select 70% of images as training data and the rest as validation data. We use NumPy random permutation to perform splitting with the seed set to 0.

### S-4.3 Long-tailed task

**Places365LT** Places365LT (Liu et al., 2019) a synthetic long-tail derivative of Places2 dataset (Zhou et al., 2018). The image resolution is 256×256. It contains 365 scene classes with at least 5 samples each. The classes are not uniformly distributed, forming a long-tailed distribution. It contains some label noises, making classification even harder on this dataset.

## S-5 Prompt templates for tasks

| Task(s) | Prompt template |
|---|---|
| ImageNet, CIFAR100, SUN397 | "a photo of a {label}." |
| FGVCAircraft | "a photo of a {label}, a type of aircraft." |
| DTD | "a photo of a {label} texture." |
| StanfordCars | "a photo of a {label}, a type of car." |
| Food101 | "a photo of {label}, a type of food." |
| Flowers102 | "a photo of a {label}, a type of flower." |
| OxfordIIITPets | "a photo of a {label}, a type of pet." |
| EuroSAT | "a centered satellite photo of {label}." |
| UCF101 | "a video of a person doing {label}." |
| Places365LT | "a photo of the {label}, a type of place." |

Table S1: Prompts of tasks

CLIP (Radford et al., 2021) suggests utilizing a sentence template (e.g., `"A photo of a {label}."`), as input to the text decoder instead of a plain text label, due to its training data being primarily full sentences describing images. Consistent with this paper's focus, we employ a simple prompt template for each task. Most of these templates are based on CLIP's recommendations[1] and are summarized in Tab. S1.

---

[1]`https://github.com/openai/CLIP/blob/main/data/prompts.md`

## S-6   Zero-shot performances on different tasks

| Task | Zero-shot Acc (%) | Official ZS Acc (%) |
|---|---|---|
| ImageNet (Russakovsky et al., 2015) | 59.7 | 63.2 |
| CIFAR100 (Krizhevsky & Hinton, 2009) | 62.3 | 65.1 |
| SUN397 (Xiao et al., 2010) | 59.2 | 63.2 |
| FGVCAircraft (Maji et al., 2013) | 18.1 | 21.2 |
| DTD (Cimpoi et al., 2014) | 42.0 | 44.5 |
| StanfordCars (Krause et al., 2013) | 58.6 | 59.4 |
| Food101 (Bossard et al., 2014) | 82.6 | 84.4 |
| Flowers102 (Nilsback & Zisserman, 2008) | 67.9 | 66.7 |
| OxfordIIITPets (Parkhi et al., 2012) | 87.5 | 87.0 |
| EuroSAT (Helber et al., 2019) | 45.4 | 49.4 |
| UCF101 (Parkhi et al., 2012) | 60.1 | 64.5 |
| Places365LT (Liu et al., 2019) | 40.0 | / |

Table S2: Zero-shot performances of CLIP ViT-B/32 pretrained model on different tasks. "ZS" is for zero-shot and "Acc" is for accuracy. Results of column "Official ZS Acc" are taken from the CLIP original paper (Radford et al., 2021). "/" represents lack of official results.

Tab. S2 shows the zero-shot performance of our implementation in different tasks. We conjecture that the main difference of official zero-shot performances comes from the ensemble prompt trick as mentioned in CLIP (Radford et al., 2021) and randomness in dataset splits of several tasks (e.g., SUN397).

## S-7   Additional results

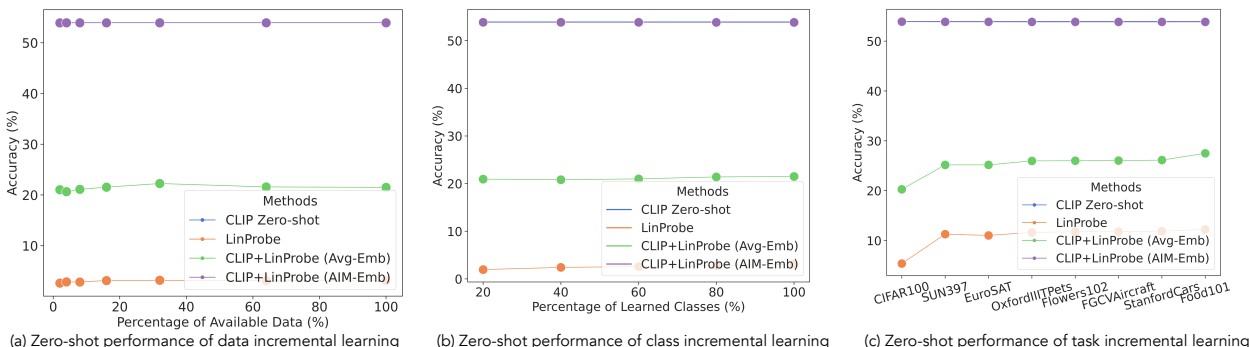

Figure S2: Zero-shot performances on data (a), class (b), and task (c) incremental learning scenarios. $x$-axis represents stages when the evaluation is performed.

### S-7.1   Zero-shot task performances of every stage

We further append the results of zero-shot tasks evaluated after each stage on the data, class, and task incremental learning in Fig. S2. Across all scenarios, AIM-Emb helps maintain the zero-shot performances on all stages.

| | Aircraft | Caltech101 | CIFAR100 | DTD | EuroSAT | Flowers | Food | MNIST | OxfordPet | Cars | SUN397 | |
|---|---|---|---|---|---|---|---|---|---|---|---|---|
| Transfer | | 87.90 | 68.22 | 45.32 | 54.61 | 71.08 | 88.86 | 59.45 | 89.07 | 64.61 | 64.05 | 69.3 |
| Aircraft | 52.45 | 87.90 | 68.22 | 45.32 | 54.61 | 71.08 | 88.86 | 59.45 | 89.07 | 64.61 | 64.05 | |
| Caltech101 | 52.48 | 96.89 | 68.22 | 45.32 | 54.61 | 71.08 | 88.86 | 59.45 | 89.07 | 64.61 | 64.05 | |
| CIFAR100 | 52.48 | 96.89 | 68.22 | 45.32 | 54.61 | 71.08 | 88.86 | 59.45 | 89.07 | 64.61 | 64.05 | |
| DTD | 52.42 | 96.83 | 81.98 | 70.32 | 54.61 | 71.08 | 88.86 | 59.45 | 89.07 | 64.61 | 64.05 | |
| EuroSAT | 52.45 | 89.92 | 81.99 | 66.65 | 95.74 | 71.08 | 88.86 | 59.45 | 89.07 | 64.61 | 64.05 | |
| Flowers | 52.51 | 90.55 | 81.93 | 66.44 | 95.74 | 54.12 | 88.86 | 59.45 | 89.07 | 64.61 | 64.05 | |
| Food | 52.48 | 90.78 | 81.94 | 67.23 | 95.78 | 65.49 | 92.25 | 59.45 | 89.07 | 64.61 | 64.05 | |
| MNIST | 52.09 | 93.95 | 81.96 | 69.73 | 94.33 | 95.69 | 92.27 | 98.59 | 89.07 | 64.61 | 64.05 | |
| OxfordPet | 52.63 | 95.10 | 82.05 | 70.05 | 95.63 | 95.69 | 92.29 | 98.58 | 92.91 | 64.61 | 64.05 | |
| Cars | 52.54 | 95.22 | 81.94 | 67.98 | 94.15 | 95.59 | 92.29 | 98.58 | 93.02 | 86.27 | 64.05 | |
| SUN397 | 52.48 | 95.56 | 81.94 | 66.91 | 95.59 | 95.59 | 92.21 | 98.60 | 93.19 | 86.15 | 81.76 | 85.5 |
| Avg. | 52.45 | 93.59 | 79.47 | 61.93 | 80.49 | 77.96 | 90.40 | 73.68 | 90.15 | 68.53 | 65.66 | 75.9 |

Table S4: Accuracy (%) of our TreeProbe (50k) model on the MTIL benchmark with order-I. Each row represents the performance on every dataset of the model trained after the corresponding task. Transfer, Avg., and Last metrics are shown in color. We follow the same table arrangement as in ZSCL (Zheng et al., 2023).

### S-7.2   More results of the comparison to previous methods

| Method | Transfer | △ | Avg. | △ | Last | △ |
|---|---|---|---|---|---|---|
| CLIP Zero-shot | 69.4 | 0.0 | 65.3 | 0.0 | 65.3 | 0.0 |
| LwF (Li & Hoiem, 2016) | 56.9 | -12.5 | 64.7 | -0.6 | 74.6 | +9.3 |
| iCaRL (Rebuffi et al., 2017) | 50.4 | -19.0 | 65.7 | +0.4 | 80.1 | +14.8 |
| WiSE-FT (Wortsman et al., 2022a) | 52.3 | -17.1 | 60.7 | -4.6 | 77.7 | +12.4 |
| ZSCL (Zheng et al., 2023) | 68.1 | -1.3 | 75.4 | +10.1 | 83.6 | +18.3 |
| KNN | 69.3 | -0.1 | 72.7 | +7.4 | 78.4 | +13.1 |
| LinProbe | 69.3 | -0.1 | 77.1 | +11.8 | 86.0 | +20.7 |
| TreeProbe (50k) | 69.3 | -0.1 | 75.9 | +10.6 | 85.5 | +20.2 |

Table S3: Comparison of different methods on MTIL in Order I from ZSCL (Zheng et al., 2023). KNN, LinProbe , and TreeProbe (50k) are complementary methods with AIM-Emb as the fusing approach.

We also supplement the results obtained by KNN and LinProbe while comparing to previous methods in Tab. S3. As shown, all of the approaches of complementary systems with AIM-Emb achieve good Transfer, reiterating the effectiveness of AIM. LinProbe excels at all metrics with the cost of efficiency, which is predictable from the results shown in our main manuscript.

To give more details on the accuracies we achieve on every task under all stages for TreeProbe (50k), we follow ZSCL (Zheng et al., 2023) to give all numbers in Tab. S4 for further reference.

## S-8   Additional ablation experiments

**Effect of different fusing operations.** We describe several forms of the fusing operations in Sec. 3.3, including: Avg-Prob, AIM-Prob, Avg-Emb, and AIM-Emb. We compare using TreeProbe (50k) under the task incremental learning scenario. Fig. S3 shows the results on target and zero-shot tasks. The figure shows that AIM-Emb and AIM-Prob have similar performance on target tasks, surpassing the other two

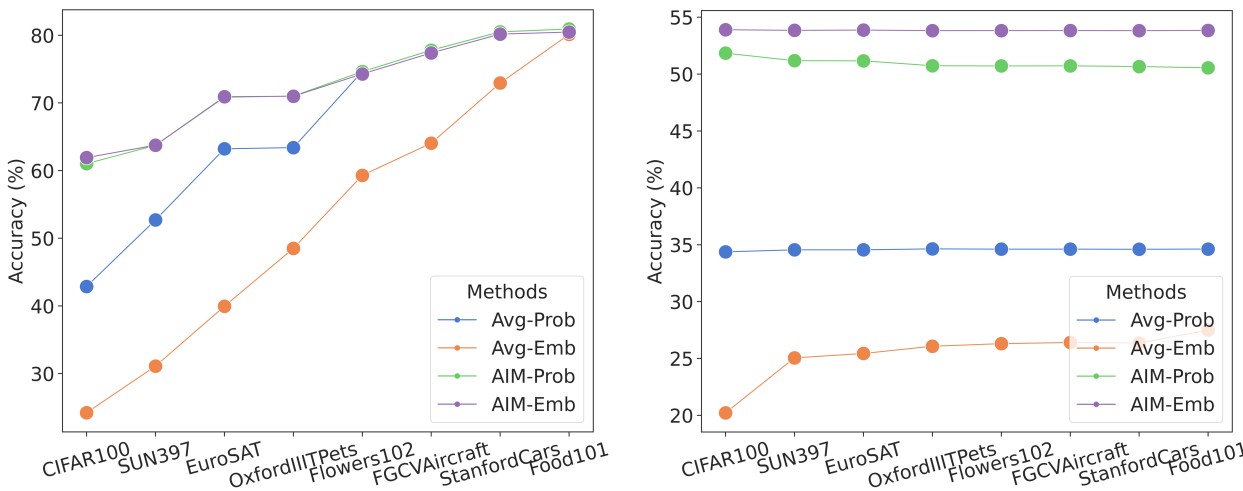

Figure S3: Target (left) and zero-shot (right) accuracies of different fusing operations.

fusing operations. Combined with the zero-shot performance, the results suggest the effectiveness of AIM in adaptively choosing the better prediction model. The probabilistic prediction is better than the embedding prediction when performing averaging in both target tasks and zero-shot tasks. But when combined with AIM, the embedding version has a reasonably better performance in zero-shot tasks. Therefore, we choose AIM-Emb as the default fusing operation.

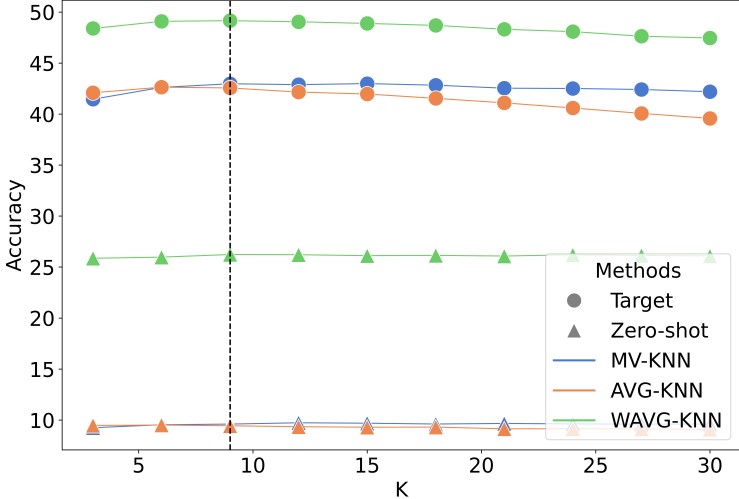

Figure S4: Results of different versions of KNN on target tasks after finishing all stages under the task incremental learning scenario. We further ablate on $k$ selection so choose $k$ as the $x$-axis. The vertical dashed line represents $k = 9$.

**Different versions of KNN and choice of** $k$**.** As indicated in Sec. 3.2, we can get the embedding by taking the one attached to the most likely label (`MV-KNN`), or averaging the text embeddings of the $k$ nearest neighbors (`AVG-KNN`), or performing a weighted-averaging over the embeddings of all $k$ nearest neighbors where weights come from the similarities between the image embedding and the $k$ neighbors' image embeddings (`WAVG-KNN`). We also compare these approaches under different $k$s to further choose a suitable $k$ for experiments. As indicated in Fig. S4, from the curve, $k = 9$ gives reasonable performances of all approaches on both the target and zero-shot tasks. Compared to `AVG-KNN`, larger $k$ is more beneficial for `MV-KNN` since larger $k$ is more stable for `MV-KNN`, and it is more likely to include more mismatches from the nearest neighbors for `AVG-KNN` .

From the plot, we can clearly read that `WAVG-KNN` is consistently better than `AVG-KNN` and `MV-KNN` across different $k$s, making it our default option of the prediction approach for fast learning system.

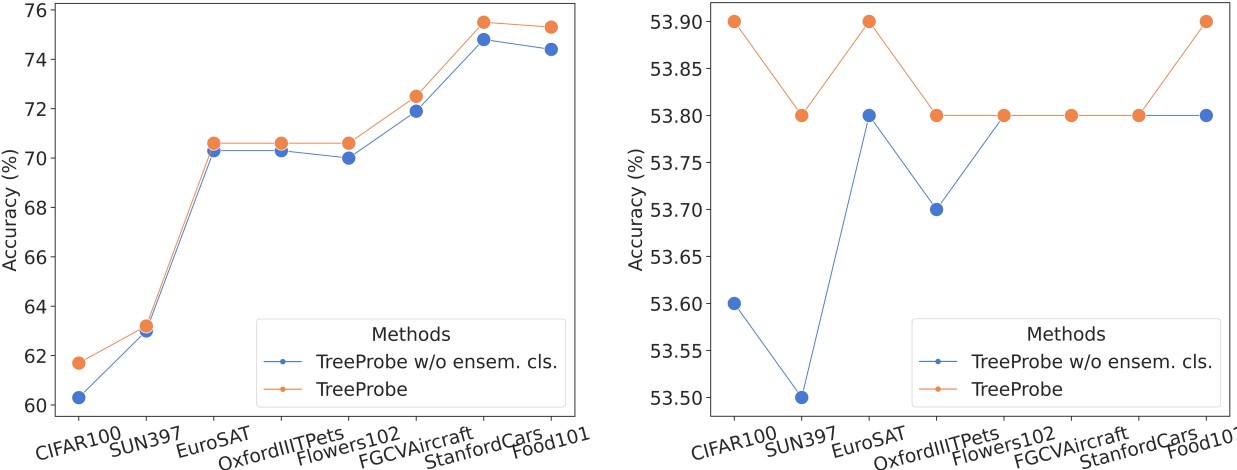

Figure S5: Target (left) and zero-shot (right) task performance comparison w.r.t. ensemble classifiers. "TreeProbe w/o ensem. cls." is the version of TreeProbe by finding the cluster most similar to the input and using the corresponding classifier to predict labels. The reported accuracy is the average across tasks after cumulatively receiving training data from the datasets shown on the $x$-axis.

**Effect of ensemble classifiers in TreeProbe inference.** Referencing Sec. 3.2, we observe that ensemble predictions from multiple classifiers associated with $k$ retrievals slightly enhance performance. Fig. S5 presents these results under the task incremental learning setting for eight target tasks. Our final model, TreeProbe, is better than its variant without the ensemble classification function in both target and zero-shot performance. The additional inference cost for ensemble predictions is negligible, so we choose it as the default setting for its better performance.

## S-9   Evaluation on long-tailed classification

| Method | CLIP Zero-shot | KNN | LinProbe | TreeProbe | KNN* | LinProbe* | TreeProbe* | PaCo | RAC |
|---|---|---|---|---|---|---|---|---|---|
| Accuracy (%) | 40.0 | 35.5 | 37.0 | 30.5 | 40.4 | 42.7 | 41.3 | 41.2 | 47.2 |

Table S5: Comparison of long-tailed classification on Places365LT (Liu et al., 2019). * means + AIM-Emb. For this experiment, we use CLIP ViT-L/14@336px as the backbone network.

In long-tailed classification, some test labels are rare or unobserved in training, so blending exemplar-based models with consolidated models can be beneficial, as shown by (Long et al., 2022). To accommodate this setting, we adjust our AIM-Emb method by considering the 2/3 rarest labels as not being present in the exemplar set and the remainder as being present, to calculate $\mathbf{v}_{\text{out}}$. In this experiment, the node capacity of TreeProbe methods is 10k. In Tab. S5, we present results on the Places365LT dataset (Liu et al., 2019). Our AIM-Emb method with LinProbe and TreeProbe outperform the zero-shot baseline. We also compare to PaCo (Cui et al., 2021) and RAC (Long et al., 2022), which are specifically designed for long-tail classification. PaCo incorporates learnable class centers to account for class imbalance in a contrastive learning approach. RAC trains an image encoder augmented with label predictions from retrieved exemplars. Although not specifically designed for long-tailed classification, our method performs similar to PaCo, but RAC performs best of all.

## S-10   Detailed time analysis

Tab. S6 compares the time complexity and actual times of three prediction approaches in a rapid learning system: KNN, LinProbe, and TreeProbe. Please refer to Sec. S-2 for software and hardware environments for this comparison. The actual times are calculated in a task incremental learning scenario, with $k$ set to 6.

KNN has a constant time complexity for both training and inference, with actual times of 9.8 and 416.1 seconds respectively. LinProbe has linear training time complexity and constant inference time complexity. The actual training time is considerably long at 30971.4 seconds ($\sim$8.6 hours), and the inference time is 449.1 seconds. Our most recommended algorithm, TreeProbe, has logarithmic training time complexity and linear inference time complexity related to the number of retrieved classifiers ($k$). In practice, it exhibits significantly shorter training time than LinProbe at 2082.0 seconds ($\sim$0.6 hour) while having a slight increase in inference time. The table illustrates that TreeProbe strikes a balance between accuracy and efficiency, being more accurate than KNN and more efficient than LinProbe. Note that numbers may vary depending on software and hardware situations. This number is collected from the same PC we used to run all experiments.

| Methods | Train time | Inference time | Actual TT (s) | Actual IT (s) |
|---|---|---|---|---|
| KNN | $\mathcal{O}(1)$ | $\mathcal{O}(n)$ | 9.8 | 416.1 |
| LinProbe | $\mathcal{O}(n)$ | $\mathcal{O}(1)$ | 30971.4 | 449.1 |
| TreeProbe | $\mathcal{O}(\log\ n + \psi)$ | $\mathcal{O}(k)$ | 2082.0 | 614.1 |

Table S6: Time analysis of different prediction approaches in rapid learning system, namely KNN, LinProbe, TreeProbe. Here, $k$ represents the number of retrieved classifiers. "TT" means training time while "IT" means inference time. Actual train and inference time is calculated with task incremental learning scenario by summing up time spend on training and inference across all tasks. For TreeProbe, $k = 6$. $n$ is the total number of exemplars that are stored, and the log $n$ is due to cluster assignment, which is negligible compared to $\psi$, the capacity of each cluster node.

