# OpenReview forum: "Continual Learning in Open-vocabulary Classification with Complementary Memory Systems"
_TMLR — Accepted by TMLR_

### Review · Reviewer_LQD8 · 2024-07-23

**Summary Of Contributions:**

The paper introduces a novel method for open-vocabulary continual learning by leveraging inspiration from the complementary learning systems observed in human cognition.

The approach integrates predictions from a CLIP zero-shot model with an exemplar-based model, using zero-shot probability estimates to determine class membership.

The author introduces the tree probe method, which uses hierarchical clustering for nearly constant training time and better accuracy than KNN approaches.

The authors conduct numerous experiments to validate the efficacy of their proposed method.

**Audience:**

Yes

**Broader Impact Concerns:**

The authors discussed broader impact concerns in their paper. Moreover, this article primarily considers common publicly available image datasets and does not involve personal privacy, thereby presenting no ethical issues.

**Claims And Evidence:**

Yes

**Requested Changes:**

**Problem 1:** Why do the LinProbe and CLIP + LinProbe (Avg-Emb) curves overlap in Figure 2(a) and (b)? Could the authors provide a more detailed explanation?

**Problem 2:** Additionally, it is recommended that the authors include more baseline methods for comparison and use a column to specify whether a vision-language model is used in the table.

**Problem 3.**  When set $\alpha=p\left(y \in \mathbf{Y}_e \mid I\right)$, equation(3) is different from equation(5).   Could the authors give more explanations how to get AIM(Adaptive Instance Marginalization) equation(5) from equation(3)?

**Strengths And Weaknesses:**

# Strengths

The concept of extending continual learning to open-vocabulary learning is rather interesting. This approach broadens the scope of continual learning, making it applicable to a wider range of vocabulary.

The combination of a visual-language foundation model like CLIP with an exemplar model vividly simulates the human cognitive process. By leveraging the strengths of both models, the authors have created a robust system.

The tree probe method stands out as an impressive innovation. It integrates incremental hierarchical clustering into continual learning, thereby enhancing efficiency.

The authors conducted extensive experiments to validate their proposed method. These experiments were carried out under various settings, including data incremental, class incremental, and task incremental scenarios.

# Weaknesses
The improvement of the model appears to stem primarily from the adaptation of the CLIP model, resulting in performance only slightly better than that of ZSCL, another model utilizing CLIP.

The reliance on a frozen CLIP model constrains the potential performance gains after fine-tuning.

Additionally, this method faces challenges in generalizing to more complex tasks, such as semantic segmentation, limiting its broader applicability.

---

> ### Author Response · Authors · 2024-07-25
> **Response to the initial review of Reviewer LQD8**
>
> Thank you for your timely review and valuable comments. We appreciate your thinking that we broadened the scope of continual learning, worked on a rather interesting problem, created a robust system, made an impressive innovation, and conducted extensive experiments. Next, we provide our responses on one weakness and all questions.
>
> **Performance only slightly better than ZSCL**:
> While we agree that our accuracy is only 1-2% better than ZSCL (Table 1), we would like to point out other advantages of our method:
> 1. Almost no forgetting: 0.1% for ours vs. 1.3% in ZSCL
> 2. Consistency: We use the same hyperparameters across all tasks, unlike ZSCL.
> 3. Efficiency: Our method updates by training linear classifiers on a subset of the data, while ZSCL requires more GPU memory, storage, and computation to update the entire model.
>
> **Problem 1**:
> Avg-Emb ($ 0.5 \mathbf{v}_e + 0.5 \mathbf{v}_I$) averages the label embeddings ($\mathbf{v}_e$) from LinProbe (linear classifier weights) and the original CLIP embedding  ($\mathbf{v}_I$). The linear classifier tends to contribute much more to the logits in this case, so that Avg-Emb and LinProbe make similar predictions when at least some of the labels are in the exemplar set.  Our proposed AIM-Emb provides a more effective way to combine, and we  compare to additional fusion methods in supplemental (Fig. S3).
>
> **Problem 2**:
> We agree with your statement that the paper contains "extensive experiments to validate the proposed method" and, given the large number of results that are already present in the main paper and supplemental, are hesitant to include more without cause. Could you please let us know which additional baselines are critical to include and why? If additional baselines are not critical, we respectfully request to allow the current validation to be sufficient.
>
> **Problem 3**:
>  The first term in Eq.5 represents the probability of a label that is within the exemplar set, i.e. that the label is in the exemplar set and of the particular label in that set. This term uses predictions from both the exemplar and zero-shot models.  The second term represents the probability that the label is not in the exemplar set and of the particular label, using only the zero-shot model (since the exemplar model is not applicable in that case). The fraction in the first term represents the probability of the label **given that it is in the exemplar set**, based on the CLIP and LinProbe predictions.  We assume that the CLIP and LinProbe models make independent predictions, and the CLIP model encodes uniform label prior (since the CLIP embedding is unit-normalized).  The denominator ensures that the total probability within the exemplar set sums to 1. We will explain the motivation of the equation more clearly in a revision.

---

> > ### Author Response · Authors · 2024-08-17
> > **Additional experiments in the revised manuscript**
> >
> > Dear reviewer,
> >
> > We want to remind you that we've provided additional comparisons with ZSCL by running it on our benchmark and supplemented additional comparison of training efficiency. Those results under task incremental learning scenario and flexible inference are in Fig. 5 (c)(d), and the training efficiency comparison is in Tab. 1. We encourage you to check the revised manuscript for more details. In general, our approach is better than ZSCL in performance, training efficiency, and consistency in training hyperparameters of different learning scenarios.
> >
> > Thank you!

---

### Review · Reviewer_avc7 · 2024-08-04

**Summary Of Contributions:**

This paper considers the problem of continual open-vocabulary classification. To solve this problem, the authors propose a simple, yet effective approach, by combining the predictions of CLIP baseline and examplar-based model. Experiments are conducted on various datasets, showing the improvements of the proposed method over previous methods.

**Audience:**

Yes

**Broader Impact Concerns:**

The authors have already discussed broader impact.

**Claims And Evidence:**

Yes

**Requested Changes:**

Please solve the Weaknesses.

**Strengths And Weaknesses:**

**Strengths**

- This paper studies an interesting problem, i.e., continual open-vocabulary classification. A comprehensive analysis is provided in this task.

- The paper is well-written and easy to follow.

- The paper proposes a simple yet effective method, which achieves consistent improvement over CLIP baseline.

- Extensive experiments are provided, demonstrating the effectiveness of the proposed method.



**Weakness**

- In the introduction, it would be better to show a figure to illustrate the motivation of the method, the task definition of continual open-vocabulary classification, as well as the benefit of the proposed method.

- Figure 2 is hard to understand. For example,  what is AIM in the figure? Examplars in different classes should be in different colors.

- Using CLIP as incremental learning is not novel. This paper lacks of comparison of more recent CLIP-based methods.

- Instead of combining with the proposed examplars-based method, I think it can also be combined with others, such as prompt-based methods. To demonstrate the effectiveness of the proposed method, the authors should compare with the combinations with other types of incremental learning methods.

- I think there may be a drawback when there is a domain gap in the testing data. For example, when evaluating the generalization ability on ImageNet-V2-Sketch-A-R, the examplar-based module will produce low results on these target domains. In this situation, combining with examplar-based module may bring negative impact to the whole system.

---

> ### Author Response · Authors · 2024-08-17
> **Author responses**
>
> We appreciate your decent work in reviewing our paper and provide insightful comments. Besides, we thank you for  thinking our problem is interesting, the method is simple and effective, and the experiments are extensive. We also want to kindly remind you that we provide a revised manuscript with supplemented experiments and details and it is a good source of information for our following responses to your concerns/questions:
>
>
> **Create a figure to show motivation** Thanks for bringing this to our attention. We’ve made a figure demonstrating the motivation of this work in the revised manuscript. We hope this figure makes it more intuitive to understand our motivation.
>
>
> **Figure 2 is hard to understand** Thank you for your suggestions to make the figure more clear. We’ve followed your suggestions to make changes to the figure in the revision.
>
>
> **Using CLIP as incremental learning is not novel and lack of comparison to recent CLIP-based methods** We are not aware of many methods built upon CLIP for continual learning except for ZSCL and a new paper [1] which mainly offers a new dataset for continually training CLIP for the image-text alignment task. The approach of [1] is to use the latest checkpoint and then rehearse over all old data, sharing a similar idea to ZSCL. We, therefore, perform a thorough comparison to ZSCL by implementing it under our benchmark and put the performance comparison under task incremental learning scenario and flexible inference in Fig. 5 (c)(d), and the training efficiency comparison in Tab. 1. We encourage the reviewer to check the revised manuscript for more details. In general, our approach is better than ZSCL in performance, training efficiency, and consistency in training hyperparameters of different learning scenarios.
>
>
> **Combine with other methods** We thank you for this suggestion. Our main technical contribution of this paper is to introduce efficient exemplar models such as TreeProbe to be used in conjunction with a consolidated CLIP model for continual open-vocabulary classification. We hold that our extensive experiments are already valid to support these contributions. Exploring more combinations of methods to improve the exemplar model is not the goal of this paper and should be deemed as future work. Therefore, we respectfully request that our current evaluation is sufficient.
>
>
> **Evaluating the generalization ability on ImageNet-V2-Sketch-A-R** We agree such a concern can be valid. This requests a more flexible and intelligent weighting method to balance exemplar models and the zero-shot model. This is an exciting future research direction for us.
>
>
> [1] Garg, Saurabh, et al. "Tic-clip: Continual training of clip models." arXiv preprint arXiv:2310.16226 (2023).

---

### Review · Reviewer_FUce · 2024-08-04

**Summary Of Contributions:**

This paper presents a method that leverages pretrained multimodal embeddings for continual/incremental learning of image classification. The proposed method leverages a TreeProbe algorithm to learn a small and fast model on top of the frozen CLIP embeddings. Experiments ablated different design choices of the system and performance of the proposed system is compared with results in prior work.

**Audience:**

Yes

**Broader Impact Concerns:**

the broader impact statement is present and sufficient

**Claims And Evidence:**

Yes

**Requested Changes:**

more baselines/ablations needed (see weaknesses above)
- ablate different VLM embeddings
- some finetuning version of the encoders

improving the performance margin compared with ZSCL (Zheng et al., 2023), maybe with a different metric (speed) or tailored dataset?

it would be useful to separate evaluation of datasets released before and after CLIP

**Strengths And Weaknesses:**

strengths:

-How to quickly adapt pre-trained embeddings for downstream tasks is an important topic of frontier research.
-This work draws inspiration from cognitive science to design an architecture that resembles human cognition that has two different systems for fast and slow learning.
-The experiments contain multiple datasets and include ablations of CLIP and the exemplar learning system.



weakness:

-The proposed approach seems to assume the visual-textual representation (CLIP) contains all necessary information for future/downstream tasks since it does not update or finetune the encoders. This is a major limitation for applications with larger downstream domain gaps.

-Using CLIP and learning a smaller network on top of the pretrained embeddings for continual learning is not a novel idea itself, and TreeProbe is an incremental contribution on top of linear probe. Therefore the major contribution of this work comes from the empirical results. However, the performance gain compared with the prior work ZSCL (Zheng et al., 2023) seems marginal and therefore the contribution isn't convincing enough. It would be useful to compare the learning and inference speed with this prior work.

-At the same time, while the authors ablated multiple different versions of CLIP, it is desirable to also ablate different vision-language models (possibly stronger ones) to gain a comprehensive understanding of the usefulness of the proposed system.

-The experiments section of this paper is not easy to follow with too many different settings. It would be useful to include some qualitative results and example evaluation data.

---

> ### Author Response · Authors · 2024-08-17
> **Author responses**
>
> We sincerely thank you for taking the time to provide such a detailed review. We truly appreciate your thorough analysis and the effort you've put into sharing your thoughts. Your insights are invaluable and will help us improve our work. We also want to kindly remind you that we provide a revised manuscript with supplemented experiments and details and it is a good source of information for our following responses to your concerns/questions:
>
>
> **Using CLIP and learning a smaller network is not novel. TreeProbe is incremental.** We stand by our paper's contribution claims on page 2 that aspects of TreeProbe, our way of combining a linear model with a frozen one, and our flexible learning/inference evaluation are original.
>
>
> **Gain over ZSCL not significant, require more metrics for comparison.** Thanks for offering additional metrics for a thorough comparison to ZSCL. We follow your recommendation and perform a side-to-side comparison to ZSCL including performance of our task incremental learning scenario, flexible inference and training efficiency in Fig. 5(c)(d) and Table 1 of the revised manuscript. From the comparison, we are > 20 times faster than ZSCL (in Tab. 1) in training efficiency while achieving better target task performance especially in later stages (in Fig. 5(c)). Besides better performance and efficiency, our method is also advantageous for its consistency in using the same set of hyperparameters across different tasks, unlike ZSCL.
>
>
> **Ablate additional vision language models** While we acknowledge that experiments with additional language models could be of interest to some readers, that extends beyond the claims and scope of our paper.  CLIP continues to be the most widely used vision-language embedding, and it's not clear whether results on current alternatives will be of long-lasting interest.
>
>
> **Would be useful to include qualitative samples** Thank you for the suggestion. We’ve included a new figure offering more intuitive explanations of different evaluation settings using image samples from the involved datasets. We hope this offers better illustrations of our setup and decreases the difficulty of understanding.
>
>
> **More ablation needed for fine-tuning versions of encoders** Fine-tuning encoders is an effective approach to improve model’s performance with the cost of more training computation. In the new comparison shown in Tab. 1 of the revised paper, we illustrate that fine-tuning CLIP is around 10x slower than our approach. This can be an issue in real world applications where learning efficiency is important.
>
>
> **Separate evaluation datasets before and after CLIP** This is an interesting idea! We assume the motivation of this experiment is to show how the system generalizes to domains where CLIP fails to specialize. However, this can be tested using datasets where CLIP fails to perform well initially prior to training. Our evaluated datasets, such as EuroSAT, Flowers102, StanfordCars and SUN397, which get significant performance improvements after training, can be viewed to function similarly as the proposed benchmark.

---

### Decision · Action_Editor_46bD · 2024-09-18

**Recommendation:** Accept as is

**Comment:**

The reviewers provided a mix of positive and constructive feedback for the paper. They appreciated the extensive experiments across different incremental learning settings, noting that the method achieves a strong balance between learning speed, task performance, and zero-shot effectiveness. Reviewers also highlighted the significant 20x improvement in training efficiency compared to ZSCL as a major strength.

Overall, the paper was considered solid, though some reviewers felt that additional experiments and baseline comparisons could further strengthen its contributions. Two of the reviewers leaned towards acceptance, while one suggested exploring combinations with other methods and conducting experiments on datasets like ImageNet-V2-Sketch-A-R. This reviewer expressed concerns about the exemplar-based model's ability to handle domain gaps, which is a valid and important point for future work. Taking these factors into account, the Action Editor recommended accepting the paper.

**Audience:**

This paper may be interested to the audiences from various communities, such as machine learning, computer vision, and even NLP.

**Claims And Evidence:**

The paper, "Continual Learning in Open-vocabulary Classification with Complementary Memory Systems," proposes a method for continual learning in open-vocabulary image classification, inspired by human cognitive systems.
It combines predictions from a CLIP zero-shot model and an exemplar-based model, using zero-shot probabilities to determine whether a sample’s class is within known categories. A key innovation is the "tree probe" method, which leverages lazy learning principles for rapid adaptation to new examples, achieving competitive accuracy to batch-trained models with significantly faster learning. The approach is tested in various incremental learning settings—data, class, and task—demonstrating a strong balance between learning speed, task performance, and zero-shot generalization.

Experiments show the method provides a 20x improvement in training efficiency while outperforming previous methods like ZSCL in accuracy, especially in later stages of learning​. The experiments are extensive enough to support the the claims in the paper. And the reviewer suggested the combination with other method, as well as evaluation on more challenging  ImageNet-V2-Sketch-A-R dataset. This would take as an important future work.